# End-to-end Algorithm Synthesis with Recurrent Networks: Logical Extrapolation Without Overthinking

**Arpit Bansal**[*]
University of Maryland
bansal01@umd.edu

**Avi Schwarzschild**[*]
University of Maryland
avi1@umd.edu

**Eitan Borgnia**
University of Chicago
eborgnia@uchicago.edu

**Zeyad Emam**
University of Maryland

**Furong Huang**
University of Maryland

**Micah Goldblum**
New York University

**Tom Goldstein**
University of Maryland

## Abstract

Machine learning systems perform well on pattern matching tasks, but their ability to perform algorithmic or logical reasoning is not well understood. One important reasoning capability is algorithmic extrapolation, in which models trained only on small/simple reasoning problems can synthesize complex strategies for large/complex problems at test time. Algorithmic extrapolation can be achieved through recurrent systems, which can be iterated many times to solve difficult reasoning problems. We observe that this approach fails to scale to highly complex problems because behavior degenerates when many iterations are applied – an issue we refer to as "overthinking." We propose a recall architecture that keeps an explicit copy of the problem instance in memory so that it cannot be forgotten. We also employ a progressive training routine that prevents the model from learning behaviors that are specific to iteration number and instead pushes it to learn behaviors that can be repeated indefinitely. These innovations prevent the overthinking problem, and enable recurrent systems to solve extremely hard extrapolation tasks.

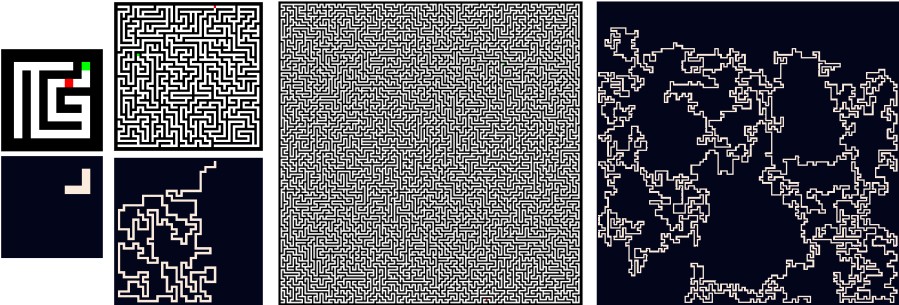

Figure 1: A 'thinking' network trained on $9 \times 9$ mazes and their solutions (left) autonomously synthesizes a scalable algorithm. By running this algorithm for longer, it reliably solves problems of size $59 \times 59$ (middle), $201 \times 201$ (right), and much larger (appendix) without retraining. Standard architectures, and even existing primitive thinking models, fail to tolerate this domain shift.

___
[*]Equal contribution.

36th Conference on Neural Information Processing Systems (NeurIPS 2022).

# 1  Introduction

Humans solve complex logical reasoning problems through *logical extrapolation* – they assemble simple logical primitives into complex strategies. For example, a person taught to prove simple lemmas can in turn prove more complex theorems simply by expending more cognitive effort.

Neural networks have achieved great success at pattern matching tasks, often exceeding human performance, but they struggle to solve complex reasoning tasks in a scalable, algorithmic way. Recently, Deep Thinking systems have been proposed as a way to represent and learn scalable reasoning processes using recurrent neural networks [22]. The word 'thinking' in this context refers to sequential processing to solve discrete/logical problems. These systems train recurrent models (networks that recycle parameters between layers) to solve reasoning problems. Unlike traditional feed-forward models, which are limited in the complexity of problems they can solve by their finite depth, the effective depth of recurrent models can be expanded after training simply by iterating the recurrent unit for longer.

When trained properly, thinking systems learn scalable algorithms for solving classes of problems. After training to solve small/easy problem instances with few recurrent iterations, the algorithm is then extended to run for more iterations at test time. In doing so, the system can achieve algorithmic extrapolation, solving problems of greater difficulty than those in the training set.

To date, the level of algorithmic extrapolation observed in thinking systems has been quite modest. For example it has been demonstrated that a system trained on $9\times9$ mazes can extrapolate to solve a $13\times13$ maze. These systems fail to achieve greater extrapolation because of a problem we call *overthinking*; recurrent systems, when extended too far outside their training regime, often deteriorate and fail to produce interpretable outputs.

In this work, we design purpose-built neural architectures and specialized training loops to make it possible to train systems that do not suffer from overthinking and instead converge to a fixed point when iterated for thousands of iterations. By doing so, we are able to build thinking systems that exhibit extreme algorithmic extrapolation behaviors, and leap from solving small/simple training problems with tens of iterations to solving large and complex problem instances at test time using thousands of iterations.

We experiment on benchmark problems for measuring extrapolation behavior. These tasks include computing prefix sums, finding optimal solutions for two-dimensional mazes, and solving chess puzzles [21]. Our architectures and training routines significantly outperform existing methods on all tasks in the benchmark suite. Additionally, we demonstrate that iterative methods for these reasoning problems are susceptible to overthinking, a problem that is overcome by our new architectures and training loops.

Our contributions can be summarized as follows:

- We provide a recurrent architecture for algorithmic extrapolation, in which the problem input is concatenated directly to the feature stack of certain layers in the recurrent thinking module. This prevents the problem instance from being forgotten if deep features become noisy, corrupted, or lossy.

- We develop a new training routine that incentivizes recurrent networks to make incremental improvements towards a solution, improving the feature representation after each iteration. This training process removes information about how many times the recurrent module has been applied, preventing the network from learning iteration-specific behaviors and instead allowing models to learn scalable behaviors that can be iterated indefinitely for extrapolation.

- We analyze the overthinking problem and show that our models overcome this phenomenon. In some cases, the algorithms learned by the proposed networks appear to be capable of solving problems of arbitrary size, despite training only on very small problem instances.

Our improvements in performance on the easy-to-hard benchmark datasets can be categorized in several ways. First, our models yield uniformly higher accuracy across the most difficult tasks used for testing in previous work. Second, we show that our models can extrapolate to much harder/larger examples than are considered in previous work, where the prior methods generalize poorly, if at all. Lastly, we show that our models do not forget solutions in settings where previous models overthink.

## 2 Related work

Preliminary work [22] shows that simple recurrent architectures, when trained to solve various reasoning problems, can exhibit algorithmic extrapolation while their feed-forward counterparts cannot. In this section, we contextualize this approach amongst prior work on algorithm learning, adaptive neural models, and logical extrapolation.

**Algorithm learning** describes models that learn scalable processes from data. Early works on this topic study the ability of recurrent neural networks (RNNs) to process input strings of arbitrary lengths [7, 20]. More recent work by Graves et al. [9] introduces neural Turing machines designed to mimic programmable computers, and Kaiser and Sutskever [13] propose a parallel version inspired by massively parallel graphical processing units. These methods, and various improvements to them, show promising results on bit string to bit string tasks, including copying inputs and adding integers, and even demonstrate the ability to generalize from shorter training strings to longer ones at testing [9, 13, 6]. Since they are based on classical RNNs, however, the amount of computation they perform is directly linked to the length of the input string, which prevents them from executing more or less computation independently of the input size (or corresponding to the difficulty of the problem). Moreover, classical RNNs are often trained incrementally to produce one bit at a time, rather than synthesizing an algorithm for solving an entire problem end-to-end. This makes it difficult to apply them to problems where the solution cannot be decomposed into incremental parts (e.g., chess).

Constraint satisfiability problem (CSP) solving networks disentangle the amount of work from the input size [24]. Specifically, message passing neural networks can execute more passes to solve harder CSPs [24]. These systems are specific to the problem of constraint satisfaction for boolean expressions, but they are an early demonstration of scalable algorithmic behavior.

**Adaptive neural networks** are designed to expend varying amounts of computation on different inputs, thus overcoming the limitation of classical RNNs. Self delimiting neural networks use one neuron to determine when to stop updating the hidden state in RNNs, and in doing so, they perform more or less computation for each token in an input sequence [19]. Adaptive compute time (ACT) is an algorithm that provides RNNs with a halting unit, which estimates the probability that computation should continue. This algorithm penalizes 'ponder time' during training to encourage the network to solve problems quickly [8]. Eyzaguirre and Soto [5] exhibit strong performance on visual question answering by introducing a differentiable version of ACT. Adaptive transformer-based language models also exist, notably Universal Transformers and Depth-Adaptive Transformers, which utilize ACT to determine the work required for each input [3, 18]. Similarly, iterative residual networks like NAIS-Nets repeat blocks in stages and perform well on image classification [2]. All of these works test their methods *in-distribution*, i.e. where the training and testing data are sampled from the same distribution; logical extrapolation outside the training domain is not considered.

**Logical extrapolation** describes the task of generalizing to test sets which comprise more computationally complex samples than the training data. Nuamah [16] claims that "neural network models with end-to-end training pipelines ... cannot, on their own, successfully perform algorithmic reasoning," and instead proposes a hybrid hand-crafted and learned approach. Similarly, Palm et al. [17] propose recurrent relational networks, which operate on graphs by iteratively passing messages. They also claim that classical architectures lack the inductive bias to reason about relationships between objects. Several recent works call these claims into question. Schwarzschild et al. [22] employ recurrent networks based on weight-sharing architectures, which can be made deeper at test time independent of the input size. These systems exhibit logical extrapolation behavior in several domains. More details on these methods are discussed in Section 3 as our algorithms build on these directly. Banino et al. [1] reformulate the halting unit in ACT leading to probabilistic RNN models with improved performance called PonderNet, and importantly their method outperforms ACT on logical extrapolation tasks for prefix sums.

The 'thinking' systems proposed by Schwarzschild et al. [22] depart from classical recurrent networks for text, which learn from step-by-step supervision to produce output tokens one at a time. In contrast, 'thinking' systems autonomously synthesize a scalable algorithm end-to-end with no supervision over what each algorithmic step should do. Even more importantly, they can be applied to solve complex problems that are difficult or impossible to decompose by hand. In other words, sequence-to-sequence models, even adaptive RNNs, have a severe limitation in that the problem needs to be represented as a sequential input and that each token in the output is generated by the same function. 'Thinking'

systems, on the other hand, provide a mechanism for domains where this type of decomposition is
unnatural, difficult, or even impossible.

## 3    Methods

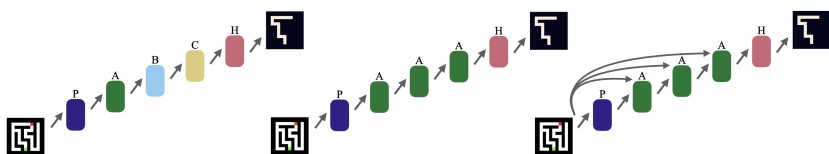

Figure 2: Architecture schematics. Left to right: A feed-forward network, a network containing three
recurrent blocks (in green) that share weights, and a recurrent network with recall.

We begin with some terms and definitions. We study networks that share weights across blocks of
layers during training. For example, instead of three distinct residual blocks, a single residual block
is repeated three times (see Figure 2 for a graphical depiction). At test time, networks trained this
way can be made "deeper" to extend their compute budget simply by repeating the block more times.
We refer to the number of layers applied in a recurrent network as its "depth," and this quantity grows
as the number of recurrent iterations increases. The number of feature maps produced by each layer
(or the number of filters in a convolutional layer) is referred to as its "width."

More formally, let $r$ be a function representing a recurrent block, e.g. a ResNet block [10], and let $r^n$
denote $n$ recurrences of that function, e.g. $r^2(x) = r(r(x))$. Let $\phi$ denote a feature map, or an output
of $r$, and let $\phi_n = r^n(x)$. We also consider an initial "embedding function" denoted by $p$, which
projects an input instance into feature space, and also a final "output head" denoted by $h$, which maps
features to outputs. A Deep Thinking (DT) network with $m$ iterations of the recurrent block can then
be expressed as follows.

$$f(x; m) := h(r^m(p(x)))  \tag{1}$$

In our systems $p$ comprises a convolutional layer followed by a ReLU, $r$ is a single four-layer residual
block, and $h$ is a set of three convolutional layers with ReLUs after the first and second. We fix $m$
during training and compute gradients for optimization by backpropagating through the unrolled
network. Then, $m$ can be increased for testing, allowing these networks to increase their processing
power and solve larger and harder problems. Below, we consider two new approaches for achieving
improved extrapolation: recall architectures and a modified optimization process.

### 3.1    Recall architectures

When humans think for a long time to solve a problem, we often stop to reread the question or review
the task at hand. We improve DT architectures to periodically recall the input exactly. We incorporate
this capability into architectures by concatenating the input problem to the features output from each
instance of the recurrent block.

Popular architectures in computer vision typically incorporate skip connections that similarly pass
information from earlier layers forward. In fact, empirical evidence suggests that skip connections,
for example in highway networks, ResNets, and DenseNets, stabilize training [25, 10, 11]. Our
architectural modification is driven by the intuition that a noisy training process creates thinking
networks that are imperfect and may leak or distort information over time as features are iteratively
fed back through the recurrent unit thousands of times. *Recall* allows the system to reproduce any
missing or damaged features, and makes it impossible to "forget" the problem being solved. To
formalize this architectural change, adding *recall* to the network can be expressed using the notation
defined above as follows.

$$f_{\text{recall}}(x; m) := h(r^m_{\text{recall}}(p(x), x)), \text{ where}$$
$$r_{\text{recall}}(\phi, x) := r([\phi, x]) \text{ and } r^m_{\text{recall}}(\phi, x) = r_{\text{recall}}(r^{m-1}_{\text{recall}}(\phi, x), x)  \tag{2}$$

Whereas the input to $r$ at iteration $k$ is usually $\phi_{k-1}$, with recall, the input to $r$ at iteration $k$ is
$[\phi_{k-1}, x]$, or the concatenation of the input with the feature map output by the previous recurrence.
We add a single convolutional layer to map the input $[\phi_{k-1}, x]$ to a feature map of the same shape as
$\phi_{k-1}$. We refer to DT networks with concatenating skip connections as DT-Recall models.

## 3.2 Promoting forward progress through optimization

We propose a training objective to encourage the system to incrementally make progress from any starting point. We do this by inputting a problem instance and running the recurrent module for some random number of iterations. We then take the output of this process, restart the recurrence in the network with these features as if iterations had just begun (discarding gradients from the initial iterations), and train the model to produce the solution after a random number of additional iterations.

This incremental training process has two benefits. First, it trains the network to continue improving the quality of partial solutions, even when they contain errors or distortions that creep in from running many iterations. Second, by choosing features from a random iteration to serve as the initial state for the training step, we discourage the network from internally counting iterations and learning iteration-specific behaviors, such as behaviors that get executed only on iteration five, for example. Rather, the network is encouraged to learn iteration-agnostic behaviors that are effective at any stage of the problem solving process.

In our implementation, we randomly sample the number of iterations used to generate a partial solution, $n$, and the number of training iterations, $k$, budgeted for the network to improve this partial solution. We then update the network's parameters to minimize loss after $n + k$ total iterations when it starts with the partial solution. This is done by detaching the recurrent module's output after $n$ iterations from the computation graph before computing

---

**Algorithm 1** Incremental Progress Training Algorithm

**Input:** parameter vector $\theta$, integer $m$, weight $\alpha$
**for** batch_idx = 1, 2, ... **do**
    Choose $n \sim U\{0, m-1\}$ and $k \sim U\{1, m-n\}$
    Compute $\phi_n$ with $n$ iterations w/o tracking gradients
    Compute $\hat{y}_{\text{prog}}$ with additional $k$ iterations
    Compute $\hat{y}_m$ with new forward pass of $m$ iterations
    Compute $\mathcal{L}_{\text{max\_iters}}$ with $\hat{y}_m$ and $\mathcal{L}_{\text{progressive}}$ with $\hat{y}_{\text{prog}}$.
    Compute $\mathcal{L} = (1 - \alpha) \cdot \mathcal{L}_{\text{max\_iters}} + \alpha \cdot \mathcal{L}_{\text{progressive}}$
    Compute $\nabla_\theta \mathcal{L}$ and update $\theta$
**end for**

---

the gradient of the loss at iteration $n + k$. The process above is an analog of truncated backpropagation through time [12], with a random starting and end point. During training, we ensure that the sum of $n$ and $k$ is less than a fixed maximum number of iteration $m$, which we call the *training regime*. The incremental loss described above is added to the standard loss computed with a full forward and backward pass through the unrolled $m$-iteration network.

The training loop and computation of the loss are given in Algorithm 1. The incremental progress term is referred to as $\mathcal{L}_{\text{progressive}}$ (or *progressive loss*), and the contribution to the loss from the fully unrolled network is denoted by $\mathcal{L}_{\text{max\_iters}}$. See Appendix A.3 for more details.

## 3.3 Datasets

We evaluate our methods on the benchmark problems available in the Python package `easy-to-hard-data`, which generates reasoning problems of various difficulties. The three problems considered are computing prefix sums, finding the optimal path in a two-dimensional maze, and solving chess puzzles. We briefly review the input and output structures for each problem and refer the reader to Schwarzschild et al. [21] for more detail, including the data generation process. Note that the architectures we consider are fully convolutional, and produce outputs of the same dimension as their inputs. Furthermore, the training and testing datasets we consider have labels of the same dimension as their inputs. Therefore, a network trained on inputs of one size can then trivially be applied to inputs of a different size.

We begin with the toy problem of computing prefix sums modulo two. The inputs and targets for this problem consist of bit strings. The $i^{\text{th}}$ bit of the target is the mod two sum of all bits prior to and including the $i^{\text{th}}$ bit of the input. We control the "difficulty" of the problem by changing the length of the bit string. Note that computing prefix sums of greater length is known to require a greater number of sequential operations [15]. All of our training is done on 32-bit strings and we explore the behavior of our models on longer strings, even showing strong performance on 512-bits.

The mazes we consider are two-dimensional square images where the walls are black, the permissible paths are white, and the start and end positions are denoted with red and green squares. The targets for this problem are maps of the same dimension as the input maze, but with ones on the optimal path and zeros elsewhere. We make more challenging datasets by increasing the size of the mazes. All training in our work is done on $9 \times 9$ mazes. Despite this small training size, we benchmark

extrapolation performance on mazes of size $59 \times 59$, and observe models solving mazes as large as $201 \times 201$ with high success rates.

The chess puzzles we consider are mid-game boards represented by twelve $8 \times 8$ planes indicating the positions of all 12 types of pieces (there are six distinct piece classes and two colors on a chess board). The goal of this task is to find the best next move, and each target encodes this information in an $8 \times 8$ array with zeros everywhere except the origin and destination of the piece to be moved, which are populated with ones. The chess dataset is sorted by difficulty rating, as determined by Elo scores computed via human trials on Lichess.org. Problems are sorted by difficulty and we train on the first 600K easiest puzzles and test our models on indices ranging from 700K to 1.1M.

Our experiments have two categories: (I) Testing extrapolation at extreme leaps in problem difficulty/size and (II) Testing scenarios that emphasize the overthinking trap and how we avoid it. The following two sections describe each of these experiments, respectively.

## 4 Extreme extrapolation

In each problem domain, we show that our models with recall and trained with progressive loss perform best.[2] We compare our models to a baseline of DT models trained with $m = 30$ and problem-specific widths (detailed below), which are both wider and deeper than the models previously studied. However, to provide a fair comparison, we use consistent values for these parameters across our experiments since we find that all architectures benefit from additional width and depth. We also compare to feed-forward models without weight sharing of the same effective depth as a 30-iteration DT network, which are ResNets [10]. We include ablation studies lending credence to our claim that both the architecture and the loss modifications are instrumental in boosting extrapolation power. For a discussion on hyperparameter choice and training set-ups for all of our experiments, see Appendix A.4. Also, code to reproduce all of our experiments is publicly available.[3]

**Prefix Sums** Though prior DT systems show extrapolation behavior on strings with up to 48 bits, we use a 512-bit testing set to illustrate the extent of our performance improvements over prior work. We train our models on 32-bit strings with a maximum of only $m = 30$ recurrent iterations (indicated in the figures as the shaded *train regime* section), 400 convolutional filters per layer, and incremental loss weight $\alpha = 1$ when progressive loss is used.

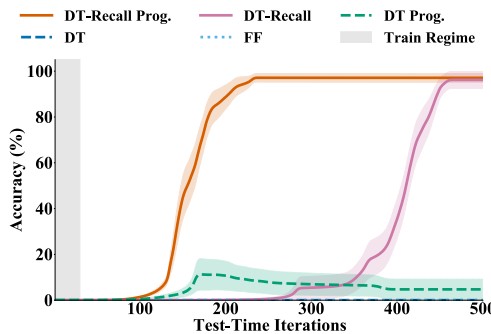 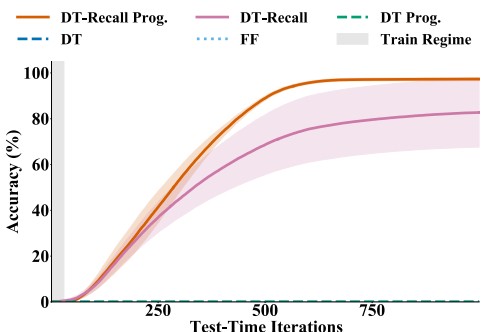

Figure 3: **Prefix sum models trained on 32-bit inputs extrapolate to 512-bit data.** The value of our recall and progressive loss is clear by how quickly and accurately our models solve this very large problem.

Figure 4: **Maze solving models trained on $9 \times 9$ inputs extrapolate to $59 \times 59$ problems.** Mazes this large cannot be solved without recall, and furthermore progressive loss leads to more accurate models.

In Figure 3, we show that without our techniques, DT networks are unable to solve very long binary strings, achieving 0% accuracy on 512-bit data, while DT-Recall models with progressive loss can solve more than 97% after approximately 200 iterations.

---

[2]Note that the DT models presented in this section use a maximum confidence exit rule for fair comparison. This exit rule marginally mitigates overthinking in the regime initially tested by Schwarzschild et al. [22]. Models with our new architecture and/or loss do not require this rule and are presented with default outputs.

[3]A PyTorch implementation of the code is available at github.com/aks2203/deep-thinking.

To better understand the individual effects of our proposed approach, we perform an ablation study. Figure 3 also makes clear that DT-Recall networks trained *without* progressive loss achieve the same final accuracy as models trained *with* it; but they require approximately twice the number of iterations to get there. Because our proposed objective succeeds in making models iteration agnostic, solutions are often found sooner than DT models that are trained to solve 32-bit problems in 30 iterations specifically. DT models trained with progressive loss outperform vanilla DT networks, however neither of these models (without recall) can solve more than a handful of 512-bit testing examples.

One other study in this domain reveals the importance of randomly setting $n$ in Algorithm 1. To show this, we modify the algorithm slightly, to always use $n = 0$, rather than randomly choosing that value. While this change seems minor, and indeed makes a small impact on the results, it does lower the accuracy. Specifically, DT networks with recall on average solve 90.27% of 512-bit test samples when trained with the $n = 0$ loss, and with randomly sampled values of $n$, as described in Algorithm 1, they achieve 97.12% accuracy. From this ablation experiment, we conclude that randomly sampling these values is, in fact, beneficial.

**Mazes** As with the prefix sum problem, we can improve performance on hard mazes by combining incremental training and recall architectures. In particular, while Figure 3 may not convince the reader that our proposed loss is critical since DT-Recall models without it perform very well, with more complex data a drastic difference emerges.

We show in Figure 4 that on the significantly harder test set of $59 \times 59$ mazes, our models exhibit strong algorithmic extrapolation, while previous methods, both feed-forward and DT systems, completely fail. Not only do our models achieve a higher peak accuracy, but they do not overthink, as can be seen by the flat spans in the curves in Figure 4. In fact, we can push these systems to their limits (and the limits of our hardware) and find that our models can solve 74% of the $201 \times 201$ mazes.[4] See Appendices A.9 and A.10 for examples of $201 \times 201$ and $801 \times 801$ mazes, which our models solve using 2,000 and 20,000 iterations, respectively (the equivalent of 10,004 and 100,004 convolutional layers in depth).

The best models in Figure 4 are DT-Recall models trained with 30 maximum iterations and a weighting in the loss of $\alpha = 0.01$. In order to see how critical recall is in overcoming the overthinking problem, we show in the same figure that without recall all models fail to extrapolate. The benefit of progressive loss is also highlighted by the fact that DT-Recall models with progressive loss achieve on average 97% accuracy while DT-Recall models without progressive loss only reach an average of 83%. Interestingly, in this domain, we find that the best model has $\alpha = 0.01$ in the loss, a much smaller weight than we use for prefix sums. More on finding the optimal value for $\alpha$ can be found in Appendix A.4.

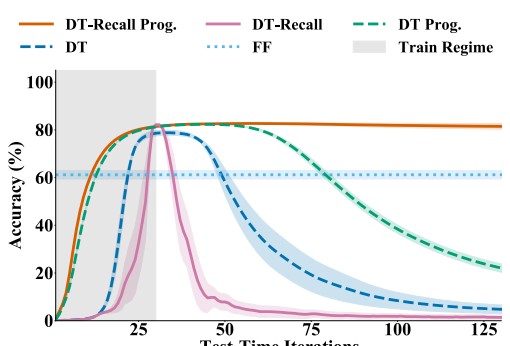

Figure 5: **Chess models trained on the first 600K easiest puzzles extrapolate to 600K-700K.** Recall and progressive loss are required to retain accuracy with many iterations.

**Chess Puzzles** We further find that using recall and progressive loss yields notable improvement on chess puzzles. In Figure 5, we see that our techniques lead to a 4% accuracy improvement compared to networks without them. We show that either removing recall or training without the incremental progress loss will hurt performance. Our best models are DT-Recall networks with 512 convolutional filters in each layer trained with a maximum of 30 iterations and a weight of $\alpha = 0.5$. Moreover, their accuracy is preserved as the number of iterations increases, while that of the other DT networks decays seriously after about 70 iterations – the sign of overthinking and the subject of the following section. Results from tests on harder puzzles are in Appendix A.5.

---

[4]The seemingly modest performance of 74% accuracy should be understood in context. Of the 100 mazes in this test set, 20 have one pixel (out of 166,464) wrong and no single maze has more than seven mistakes.

# 5 The overthinking problem

Deep Thinking networks boast impressive capabilities to solve harder problems by thinking deeper, but they are prone to overthinking. In particular, by testing models on data closer in complexity to the training data we can better compare our methods to prior work. Figure 6 shows that some recurrent networks that can perform algorithmic extrapolation may collapse entirely when performing too many iterations.

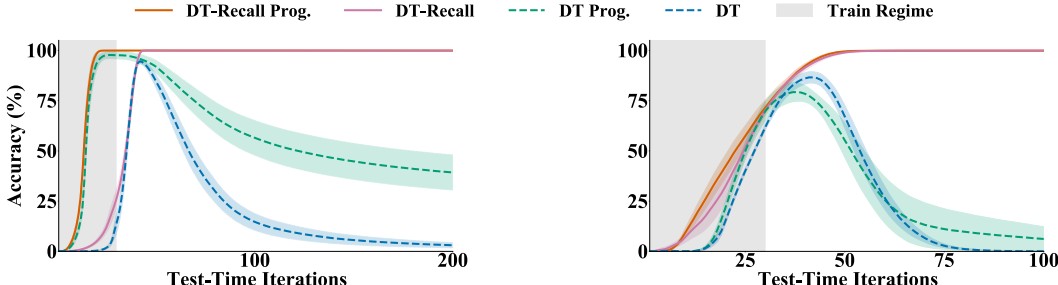

Figure 6: **Left:** Prefix sum models trained on 32-bits tested on 48-bit inputs. **Right:** Maze models trained on $9 \times 9$ mazes tested on $13 \times 13$ mazes. In these regimes, we can see that the weaker models can extrapolate to slightly harder problems, but importantly only models with recall avoid the overthinking trap.

Our models do not suffer from the sharp decline in accuracy as the number of iterations increases. In Figure 6, the decay in each dashed curve and the stability with high iteration counts of the solid curves shows that recall is critical to avoid overthinking. See Figure 5 for an example where progressive loss is needed too. The monotonic increase in accuracy with added iterations in our models is practically useful, as it allows for pre-defining a large iteration number at which to terminate the recurrence rather than having to carefully choose a stopping iteration to maximize performance while avoiding degradation.

One observation that can be made from these results is that the overthinking problem often seems to disappear when skip connections are added to provide networks with an uninterrupted view of the input. Another way to describe this is that our models seem to converge to a fixed-point solution as they iterate rather than becoming unstable. This property is desirable and using representative models trained to solve mazes, we explore how robust models are when we manipulate the features during the thinking process. In Appendix A.8, we present similar findings for prefix sums.

## 5.1 Manipulating feature maps

First, we investigate sensitivity to adding noise in the feature maps before concatenating the inputs. We examine model behavior when we add Gaussian noise (with mean zero and standard deviation one) to the features after one iteration of maze solving. Models with recall can still solve $13 \times 13$ mazes even when we perturb the features, but models without recall cannot. See Appendix A.8 for plots, tables, and further discussion.

Next, we ask whether the initial feature maps (after the projection layer $p$) carry any important information. We test this by replacing the feature maps with zeros after 50 iterations – at this point the model has solved the maze and by annihilating the features we remove all information from $p$. In this case, our DT-recall models naturally regenerate their features and recover to solve the problem again, indicating that the learned algorithm is able to find a solution using the input without the initial projection. See Figure 12 in Appendix A.8.

## 5.2 Manipulating inputs

With the notion that our models may embody a convergent process, we turn our attention to investigating how the networks determine when to stop manipulating the representation. Perhaps there is something in the feature maps output by late iterations that tells the network to stop working. We can test this in two ways. We perturb the input (which we concatenate onto the features) after some number of iterations – first subtly, then by swapping it with an entirely different example.

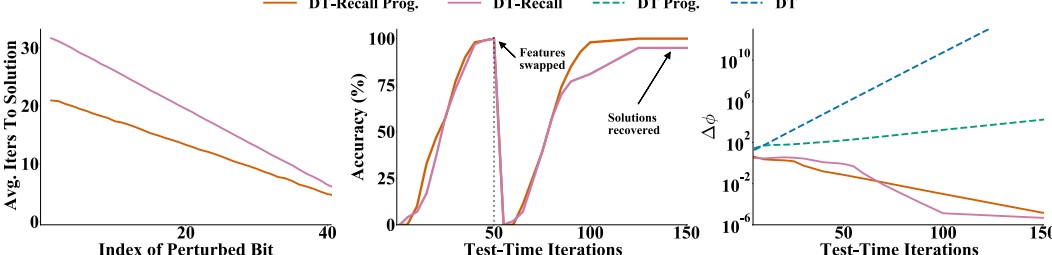

Figure 7: **Left:** How long it takes prefix sum models to recover from perturbation – recall keeps this quantity small. **Center:** Test accuracy on $13 \times 13$ mazes when features are swapped after 50 iterations. **Right:** The change in the features when solving $13 \times 13$ mazes.

To start, we flip single bits in the input string for prefix sums. We explore the response of the model to flipping bits at different indices after 50 iterations (when the models have already solved the initial 48-bit problem). The left panel of Figure 7 shows recovery time as a function of index of the flipped bit. We observe that our models can indeed recover from single bit flips, and recovery time decays linearly with index. Higher indices are closer to the end and affect fewer bits in the output than lower ones.

Similarly, with mazes, we change the input by moving the end of the maze two steps closer to the start. We use this new input concatenated with features generated after 50 iterations of solving the original maze. In this case, we see in Figure 13 (appendix) that models with recall self-correct and solve the new problem in very few iterations.

The last way we test the hypothesis that networks are continually comparing their solution to the problem instance is by partly solving Maze "A," and then swapping the features with those obtained from 50 iterations of trying to solve Maze "B." In other words: if we concatenate the input problem (A) with the features from iteration 50 corresponding to a different maze (B), which maze will the network solve? Clearly, a system without recall will solve maze B. However, with recall, networks will recover and pull the features back to representing a solution for maze A. The center panel of Figure 7 shows the effect of swapping feature maps.

### 5.3 Converging to a fixed point

We can also study the convergence by measuring the change in the output at each iteration. A decreasing change in the feature maps with each additional iteration suggests that the network manipulates the representation, moving it closer to one that solves the problem, and that it will hold onto this representation (or stay nearby) once it is reached. Moreover, we seek to qualitatively categorize each model type (architecture/loss pair) as convergent or non-convergent. To do we measure the change in the solution at each iteration with $\ell_2$-norm of the difference. The right-hand panel of Figure 7 shows $\Delta\phi(n) := \|\phi_n - \phi_{n-1}\|_{\ell_2}$. We see that our models appear to converge, while DT networks without recall explode, providing another view into overthinking.

## 6 Conclusion

In this work, we improve the algorithmic extrapolation power of neural networks. We propose an architecture and a loss that lead to a gigantic generalization leap from easy training data to much more complex testing examples. Furthermore, we show that our models avoid the overthinking trap. We test algorithmic extrapolation with chess puzzles where the spacial dimension is consistent across difficulties and with maze solving and prefix sum computation where our models can extrapolate to larger problems as well. In fact, our models use more than 2,000 iterations, the equivalent of more than 10,000 convolutional layers, to solve the largest mazes we consider.

Existing methods deteriorate at such large numbers of iterations creating the need for stopping mechanisms. Even for small numbers of iterations, if the halting decision is sub-optimal, then the output may also be sub-optimal. Because our models have the desirable property (for an algorithm) that they converge to fixed points, they run no such risk, require no learned or hand-crafted stopping rules, and can perform well at huge numbers of iterations.

Learning scalable processes and performing logical extrapolation are difficult tasks for most neural models, but with our architecture and loss, we demonstrate that huge leaps in complexity from training to testing data can be handled without overthinking. Importantly, our neural networks learn end-to-end how to perform algorithmic extrapolation. Our findings are limited to the domains we consider and further exploration into how our methods will perform in other settings is needed.

The application of machine learning systems to real world data demands effective extrapolation, even though this is often overlooked. Curated research benchmarks designed for measuring generalization (e.g. ImageNet [4]) omit a critical feature of real data in the field: there is often reason to think that data encountered after training could embody more difficult examples than those in the training set. Consider that a practitioner cannot always know *a priori* the range of complexity that exists in their domain. If we are to employ models that cannot extrapolate, we are to miss out on using our machine learning tools to solve problems harder than those for which we already have answers.

## Acknowledgements

This work was supported by the ONR MURI Program, The National Science Foundation DMS-1912866, and the Office of Naval Research (#4720008163).

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
