# OpenReview forum: "End-to-end Algorithm Synthesis with Recurrent Networks: Extrapolation without Overthinking"
_NeurIPS.cc/2022/Conference — NeurIPS 2022 Accept_

### Official Review · Reviewer_Z4mS · 2022-07-11

**Rating:** 7
**Confidence:** 4
**Soundness:** 3 good
**Presentation:** 3 good
**Contribution:** 4 excellent

**Summary:**

This paper studies recurrent networks, architectures that iterate over shared layers. Previous research by Schwarzchild et al. (ref 22 in the paper) has shown that for three specific tasks, these networks could extrapolate to larger problems than those seen at training, by being given "ponder time", i.e. more iterations at inference than at training.

The authors note that as ponder time gets large, extrapolation accuracy drops, a phenomenon they call "overthinking". In practice, overthinking prevents the model to extrapolate to problems much larger than those seen at training.

The authors propose two improvements to the original architecture:
- recall: repeating the original problem at every iteration, by concatenating it to the input of the shared layer
- progressive training: instead of computing the loss over m iterations, compute it over k successive iterations between n and n+k ( n and k randomly sampled). This amounts to randomizing the number of iterations at train time, and limiting the back propagation of gradients to the last k layers (an approach reminiscent of GNN).

On the three problems considered, they show that recall and progressive training prevent overthinking, and allow the model to generalize to much larger problems than seen at training.

**Questions:**

**Sec. 3.1**  The recall model is defined as phi(k) = r(phi(k-1), x), (phi(k) the output of iteration k, x the input sequence). Since the input of the first share layer is p(x), it would make more sense to use r(phi(k), p(x)). Have you tried this? What is the rationale for r(phi, x)?

**Sec. 3.2** Progressive training amounts to preventing the gradients to back-propagate fully. The experiments suggest that whereas it does not improve extrapolation (recall does most of the work), it speed up inference (in the sense that extrapolation is faster). Could such a principle be applied to regular deep networks, i.e. without shared layers?

What is the benefit of having the loss a weighted sum of total (m loops) and progressive (k loops) loss? Couldn't the same effect be achieved by sampling (with probability alpha) examples for which the loss would be computed for the whole m steps, and others where it would be computed on the k last? Would there be an advantage in varying alpha as training proceeds (as a form of curriculum learning)?

**Architecture** Both input and output layers (p and h) are convolutional. Also both the prefix sum and maze problems are equivariant with respect to translation, which makes conv-nets a natural choice of embeddings. How dependent on the choice of convnets are your findings? e.g. what happens if you change p (or h) to a linear layer ?

Have you done an ablation on m? in other words, could your extrapolation results allow one to reduce the number of iterations during training (hence saving memory when problems become very large)?

**misc.**
Lines 213-214 : zeroes or ones, or black and white?
Line 316 : should be chess, not prefix sums?

**Limitations:**

A discussion about the tasks for which this technique can be applied would reinforce the significance of the paper. What is the importance of translational invariance? Is the technique limited to problems that can be solved to algorithms with little or no memory (to compute the next term of the prefix sum, you only need the previous result, to choose the next move in a maze, you need limited information about your previous moves)?

**Strengths And Weaknesses:**

The paper is very well written, and the architecture clearly described. The experimental results are compelling: on the binary integration problem, the model extrapolates to sequences 16 times longer, on the maze problem, to arrays 500 times larger. This is a novel idea, and a significant result on a very important subject. It makes no doubt to me that this paper should be accepted.

The paper could be made **much stronger** by improving the experiments. In particular,
- the binary integration problem is very simple: it is a first order recurrence y[n] = y[n-1] + x[n] mod 2  (x and y the input and output sequences). Experimenting with larger order (e.g. order two, in Fibonacci fashion) and/or larger moduli would be interesting.
- as far as extrapolation is concerned, the chess results are inconclusive: both figures 5 and 9 suggest that 30 iterations are enough to achieve best performance. The authors note that recall helps achieve a slightly better accuracy, but this is not a case of Deep Thinking.

---

> ### Author Response · Authors · 2022-08-01
> **Individual Author Response: Z4mS**
>
> We thank the reviewer for the supportive review and thoughtful feedback on our work. We address the questions below:
>
> >Experimenting with larger order (e.g. order two, in Fibonacci fashion) and/or larger moduli would be interesting.
> >
> We agree with the reviewer that experimenting with higher order binary sequence tasks would be very interesting to try. We will try to include some preliminary results on the Fibonacci sequence task in the camera-ready version of the paper, and hope more thorough exploration of this will be done in future work.
>
> >The recall model is defined as phi(k) = r(phi(k-1), x), (phi(k) the output of iteration k, x the input sequence). Since the input of the first share layer is p(x), it would make more sense to use r(phi(k), p(x)). Have you tried this? What is the rationale for r(phi, x)?
> >
> The rationale for the recall architecture is to keep the original input as a “reference” for each algorithmic step. In the research leading up to the submission, we did try to concatenate the *projected input* as a form of recall. We found that using the unmodified input was more effective than using the projected version and seemed more in line with the general intuition behind recall.
>
> >Progressive training amounts to preventing the gradients to back-propagate fully. The experiments suggest that whereas it does not improve extrapolation (recall does most of the work), it speed up inference (in the sense that extrapolation is faster). Could such a principle be applied to regular deep networks, i.e. without shared layers?
> >
> We agree that this is an interesting question and would be a good avenue for future work as well.
>
> >What is the benefit of having the loss a weighted sum of total (m loops) and progressive (k loops) loss? Couldn't the same effect be achieved by sampling (with probability alpha) examples for which the loss would be computed for the whole m steps, and others where it would be computed on the k last? Would there be an advantage in varying alpha as training proceeds (as a form of curriculum learning)?
> >
> In combination with all the architectural hyperparameters, there is an enormous search space for different loss hyperparameters and loss formulations. Due to space constraints, we decided to use the general and principled formulation for progressive loss presented in the paper. We agree that curriculum learning by varying alpha during training and randomly choosing L_max_iters or L_progressive could be helpful, but we believe such modifications to make the loss more effective and/or efficient should be considered on a task-by-task basis. We believe these are rich directions for future work.
>
> >How dependent on the choice of convnets are your findings? e.g. what happens if you change p (or h) to a linear layer.
> >
> For these algorithmic tasks, it is essential for the architectures to be able to handle inputs of arbitrary size. We chose $p$ and $h$ to be convolutional layers in order to satisfy this property.
>
> >Have you done an ablation on m? in other words, could your extrapolation results allow one to reduce the number of iterations during training (hence saving memory when problems become very large)?
> >
> We performed extensive ablations of this sort on the task of solving prefix sums. Please see Figure 8 in appendix A.3 for more details.

---

> > ### Comment · Reviewer_Z4mS · 2022-08-09
> > **Thank you**
> >
> > Thank you very much for the replies.

---

### Official Review · Reviewer_fZnJ · 2022-07-12

**Rating:** 6
**Confidence:** 5
**Soundness:** 2 fair
**Presentation:** 3 good
**Contribution:** 3 good

**Summary:**

This paper proposes an approach that allows models to extrapolate to more complex problems than they observed during training, by recurrently applying the same operations many times. The paper offers two key innovations to enable this: 1) a new loss is proposed that encourages the model to converge in as few iterations as possible, rather than learning iteration-specific behaviors, and 2) a recall element is introduced that allows models to continue to have access to the original problem, which appears to prevent a phenomenon they refer to as 'overthinking' in which a model's outputs appear to drift over many iterations, even after initially discovering the correct answer. They evaluate this approach on a range of tasks that each involve generalization to more complex problems than observed during training.

**Questions:**

My questions are detailed in the 'strengths & weaknesses' section above.

**Limitations:**

I don't believe this work has any potential negative societal impact.

**Strengths And Weaknesses:**

In general I have a positive view of this work. I think the overall approach is compelling and the results are generally strong. However, I have reviewed this work before, and some of the issues raised during that discussion are addressed here, but some issues are still neglected. Here are the issues that I think still need to be addressed:
- One of the primary issues that I raised in the previous discussion is the lack of comparison with other baselines. The choice of architecture here, in which only the output from the previous stage is fed as input to the next stage, is interesting, but I still feel that the work would be improved if there was a comparison with other baselines, such as recurrent models that have a hidden state (e.g. lstm), or models with access to an external memory. It would be good to know whether this specific design decision is important to the extrapolation capability of the models.
- The progressive loss is primarily designed to encourage the models to converge quickly, without relying on having a certain number of iterations to arrive at the correct answer. A natural comparison here is some sort of penalty where loss increases as a function of the number of iterations. In the previous discussion, the authors described some experiments testing this alternative, and reported that it did not perform as well as their approach, but those experiments don't appear in the present manuscript.
- In the previous version of the paper, there was some mention of the fact that some number models were excluded from analysis because they did not reach a certain training accuracy criterion. In discussion, the authors stated that the overall results were not significantly affected by the exclusion of those models. But now there is no longer any mention of excluding models based on training accuracy, though the results appear to be the same as in the original manuscript.
- The previous version of the paper also employed something called a 'maximum confidence rule' where the iteration with the maximum confidence value was selected for testing. According to the previous discussion, this also did not affect the results much, but now there is also no mention of this selection procedure in the current paper.

So, although I'm generally positive about the work, it seems that not much has been added relative to the previous iteration of this work, and in fact some additional results that I think strengthened the paper's conclusions have now been removed. I am eager to hear from the authors about the points above.

---

> ### Author Response · Authors · 2022-08-01
> **Individual Author Response: fZnJ**
>
> We thank the reviewer for their thoughtful feedback, and we address their points below:
>
> >The choice of architecture here, in which only the output from the previous stage is fed as input to the next stage, is interesting, but I still feel that the work would be improved if there was a comparison with other baselines, such as recurrent models that have a hidden state (e.g. lstm), or models with access to an external memory.
> >
> The only problem we consider which would be applicable to traditional RNNs is prefix sums. Since we are concerned with highlighting the difference between feed-forward nets, DT nets, and DT nets with recall, we do not compare to every other algorithm to solve prefix sums (many exist and handcrafted algorithms are even provably correct). We agree that the RNN literature is rich and inspiring. In fact, in our preliminary experimentation with recurrent models, we did investigate gated layers (inspired by LSTM/GRU models), but we found that the addition made no difference and in some cases hurt training/generalization.
>
> >In the previous discussion, the authors described some experiments testing this alternative, and reported that it did not perform as well as their approach, but those experiments don't appear in the present manuscript.
> >
> We thank the reviewer for the previous discussion, which we found greatly valuable. However, due to space constraints after making other modifications from the previous version of the paper, we decided to not include any additional methods that did not improve performance. Below is some additional reasoning for this decision:
> - After further reflection on comments about the potential to save compute time by choosing a less general version of progressive loss, we identified a few conceptual problems with the approach and thus decided to only consider our original formulation for simplicity. Let's consider the case where $\alpha=1$, which happens to be optimal for prefix sums. If we take $n=0$ and reduce the expected computation costs by sampling $k$ from an interval {$1, m'$} smaller than {$1, m$}, then there will never be any forward pass through the last $m-m'$ layers. This essentially amounts to reducing the initial value of $m$, which is a hyperparameter search we have already done in the appendix (see Figure 8). For other values of $\alpha$, there is no such equivalence to a smaller $m$. However, we believe such optimizations could be performed on a task-by-task basis. We have now included this discussion in the appendix.
> - As shown in the previous discussion, preliminary tests done on models trained with a penalty for using more iterations were not effective for any of the three tasks tested in our paper. We believe a more thorough exploration of such objective functions is a rich avenue for future work, and think a brief inclusion (due to space constraints) of such results would provoke more questions than it resolves. However, based on your feedback, we have now included a paragraph in the appendix section A.3 posing different formulations of the loss as valuable future work.
>
> >In the previous version of the paper, there was some mention of the fact that some number models were excluded from analysis because they did not reach a certain training accuracy criterion...
> >
> The way we presented the training convergence criterion in the previous submission created unnecessary confusion around what is actually standard practice in the literature. For example, transformers are well known to exhibit unstable training, but it is common practice to exclude the models that fail to converge from final accuracy averages. Some readers of our work had misinterpreted the simple filtering based on training loss as some kind of trimmed mean exclusion. We aimed to avoid this confusion, but based on this feedback, we have added a *clear* statement about this in section A.5 of the appendix.
>
> >The previous version of the paper also employed something called a 'maximum confidence rule' where the iteration with the maximum confidence value was selected for testing...
> >
> After the previous submission, we found the new architecture/loss combination received *absolutely zero* benefit from the maximum confidence rule. Hence, for simplicity, all plots in the main body of this paper are tested with what we previously called the default exit rule for DT recall prog., DT prog., and DT recall. We do, however, use the maximum confidence rule for the vanilla DT nets in order to maintain a fair comparison to Schwarzchild et al. for all plots in our paper. We believe the maximum confidence rule was initially used as a way to slightly mitigate the overthinking phenomenon, but the improvement is completely negligible in comparison to our proposed architecture/loss. We have now added a footnote in the main body of the paper to clarify this point, and refer the reader to Schwarzchild et al. for details on their maximum confidence rule.

---

> > ### Comment · Reviewer_fZnJ · 2022-08-08
> > **Reply**
> >
> > Thanks to the authors for this reply. My concerns regarding the discrepancies with the previous manuscript are all addressed. I am going to increase my score accordingly, but I still think that a broader comparison with other baselines would be more informative. All of the baselines considered in this work are essentially ablations of the core proposal, so it is difficult to know how the proposed approach fares on these problems relative to different approaches. If possible, it would be good to include the results that the authors describe using gated layers, if only in the supplementary material.

---

### Official Review · Reviewer_V9gQ · 2022-07-12

**Rating:** 7
**Confidence:** 3
**Soundness:** 3 good
**Presentation:** 4 excellent
**Contribution:** 4 excellent

**Summary:**

The authors approach the task of 'algorithmic' or 'logical' extrapolation in neural networks, which is to say, the ability of a neural network to generalise patterns learned from small/simple examples in order to solve scaled up/more difficult tasks from the same overall problem, e.g., solve a large maze after seeing many small ones, or a tough chess puzzle after many easy ones.
The authors restrict themselves to problem domains in which the outputs to each task are the same size as the inputs, this allows applying convnet-based architectures to arbitrary input size without dimensionality issues.

An existing approach uses an architecture consisting of 3 convnet-based blocks: one input-processing, one recurrent, and one output-preparing. The blocks feed into each other, though the recurrent block is applied recurrently to its own input some (externally defined) number of times N before feeding to the output-preparing block. The authors revisit this architecture, noting that it a) struggles to extrapolate to especially larger examples from those seen in training, and b) may approach a task solution after some N1 recurrences, but sadly degenerate past that (i.e., giving the wrong output if asked to run for N>N1 steps). They refer to this latter issue as overthinking.

The authors suggest two modifications to the architecture, motivated by these issues. First, they suggest feeding the original input back into the recurrent net at every step, as part of its input - a "recall" architecture; and second, they suggest a careful but straightforward training regime, defining a "progressive loss" which encourages the network to output (as best it can) the correct solution at *every* step of the computation and not just the last, this pushes it to a) learn as short a solution as possible and b) stabilize on that solution once reached, avoiding the overthinking issue. Further, to prevent the network from learning iteration-specific algorithms, the authors employ BPTT when using their progressive loss (I am confused on this, see questions).

The authors evaluate the modified architecture, showing that it scales to far larger/harder problems with the modifications, and perform ablation studies demonstrating the importance of each of their modifications. They even show that their "recall" modification is truly being used for the intended purpose by the network, showing how the networks overcome perturbations in the recurrent computation to return the correct solutions for their true inputs.


**Questions:**

Please clarify the following in the paper:
1. Are all of the internal blocks in the architecture indeed convnets? Could you give (each of) their exact architectures in the paper/appendix?
2. Figure 2 suggests the original input is fed into the reccurent block (A) at every step, my intuition suggests it should be the processed input (i.e., the arrows should emanate from the 'P' block, not before it), could you clarify?
3. I am not sure why being iteration agnostic matters? What if the task really can be solved in 3 operations, regardless of input size/difficulty? I understand the value of BPTT to allow training later steps even with limited computational resources! But just don't understand the importance of being iteration agnostic.


**Limitations:**

y

**Strengths And Weaknesses:**

strengths:
- straightforward, intuitive modifications to existing work provide significant improvement in performance
- well presented, clear work (though please correct/clarify notes in comments and questions!) with convincing evaluation
- exciting task

weaknesses:
- while this is a great step for the "deep thinking" models, I don't see a lot of comparison to/discussion of other (if any) architectures that could be applied to these tasks. E.g., why not compare to a universal transformer? In this sense the work is not well contextualized.

comments, including presentation comments and typos:

0. (general) this work uses 'recurrence', but in a different way from RNNs: RNNs recur alongside an input sequence, eating one token at a time and finishing at the end of the input, this architecture simply recurs on its own state (albeit with help of the input recall) for a given number of steps, independently of the input. Given the popularity of RNNs, for clarity, I suggest this distinction is clarified early in the paper, e.g. a sentence of the sort "this work uses recurrence in a manner different from that commonly referred to in RNN literature, the architecture behaviors should not be confused"
1. line 8 "because behavior degenerates" I don't know if 'behavior degenerates' so much as iterations continue past correct stop (losing solution), or representation degenerates (also losing solution), or some other issue. Rephrase (unless you prove that indeed "behavior degenerates" and is the "cause" of the failure to scale).
2. Fig 1 caption description of 59x59 as "center" is confusing, consider just labeling the subfigures
3. line 20 "lack the ability to solve complex reasoning tasks in a scalable, algorithmic, way" again strong concrete statement, not sure I agree, soften/rephrase.
4. Eq 2 (description of f_recall, r_recall) - bad notation: r_recall takes two inputs, one of which is its own previous output, and the other the original input to the net. By construction such a function cannot be applied recurrently (r^2_recall(phi,x) unrolls to r_recall(r_recall(phi,x)), which makes no sense because the x has to be put in again). Consider instead notation similar to that used in RNN descriptions, e.g.: phi_t+1=r_recall(phi_t,x), phi_1=r(x), f_recall(x,m)=h(phi_m) (something like that - fill it out right according to your architecture. Please also make sure to properly distinguish between the original input x and the processed input P(x), so we can tell which one is being used where!).
5. line 163: "re-initialize the network" - the entire network or just the recurrent part? clarify
6. line 164: clarify here that gradient is being discarded, easier to follow that way
7. line 169-170: (too) strong statements! 1) the network may arbitrarily implement counting its iterations, it is not "prevented" from doing so, and 2) the network might still learn iteration specific behaviors, whether by luck or even directly through the SGD (given that the standard loss is also still used!). The only claim I would make here is that encouraging random iterations to be correct a) encourages a stabilizing behavior (i.e. not losing the solution once obtained) and b) encourages 'faster' (shallower) solutions (i.e. having the solution ready in the earliest iteration possible) - basically, it pushes the network to have the correct solution available in the maximum number of iterations possible.
8. Fig 3: would also be nice to see the stability of DT-recall (i.e. no prog loss) *after* reaching the solution, e.g. plot accuracy up to 1000 iterations (given that prog loss is supposed to encourage stability, and here we only just see DT-recall reach the solution but not whether it stabilizes there)
9. line 254-255: [s/of sampling/of randomly sampling], [s/We modify/To show this, we modify], [s/slighty to always/slightly, to always]
10. line 259 s/random/randomly
11. line 265 "harder" - harder than what?
12. line 293-294: explicitly mention that the best models had progressive loss (currently missing from this sentence)

---

> ### Author Response · Authors · 2022-08-01
> **Individual Author Response: V9gQ**
>
> We thank the reviewer for the extensive feedback on our work, and we have incorporated many changes as described below:
>
> >comments, including presentation comments and typos:
> >
> We found these suggestions very helpful, and have modified our paper to address them.
>
> >Are all of the internal blocks in the architecture indeed convnets? Could you give (each of) their exact architectures in the paper/appendix?
> >
> Yes, they are all convnets. See Section A.1 of the appendix for more details, and please let us know if more clarification is necessary on the architecture.
>
> >Figure 2 suggests the original input is fed into the reccurent block (A) at every step, my intuition suggests it should be the processed input (i.e., the arrows should emanate from the 'P' block, not before it), could you clarify?
> >
> Yes, the intuition expressed by Figure 2 is correct, as we indeed feed the original input into the recurrent block at every step. Please let us know if/where we can clarify the text to help avoid such confusions for other readers.
>
> >I am not sure why being iteration agnostic matters? What if the task really can be solved in 3 operations, regardless of input size/difficulty? I understand the value of BPTT to allow training later steps even with limited computational resources! But I just don't understand the importance of being iteration agnostic.
> >
> We consider algorithmic tasks where more iterations **are** indeed needed to solve larger problem instances.  In this context and with a training set consisting of problems of a fixed size, the standard cross-entropy loss will lead the network to perform iteration-specific behaviors which won't scale when more iterations are required.  For example, the network may learn to aggregate information at a fixed iteration $k$ anticipating solving the problem by that iteration, but then at test time, the model encounters much larger problem instances requiring many more than $k$ iterations and is not equipped to solve them.  Hence, the idea behind the iteration agnostic property of progressive loss is to avoid such iteration-specific behaviors.

---

### Official Review · Reviewer_nMzj · 2022-07-19

**Rating:** 7
**Confidence:** 3
**Soundness:** 3 good
**Presentation:** 3 good
**Contribution:** 3 good

**Summary:**

The aim of this paper is to extend the generalization of learning approaches on algorithmic challenges.

They introduce an architecture for algorithmic extrapolation, which amounts to providing the input to the network as input to each current layer.

They introduce a training procedure that aims to train a network to improve a partial solution in an incremental manner.  Specifically, the loss term minimizes an error term not relative to the actual input $x$ but instead starting at some intermediate value.

**Questions:**

If I understand correctly:

1. $\mathcal{L}_{\textrm{max iters}} = |\hat{y}_m - y| = |f_\text{recall}(x; m) - y|$ and:
2. $\mathcal{L}_{\textrm{progressive}} = |f_\text{recall}(x_n; k) - y|$ where $x_n = f_\text{recall}(x; n)$

Is this correct? If so, I suggest adding an expression like this to the paper.  Currently the loss terms are not fully specified.  Section 3.2 and Algorithm 1  give a procedural account of how to compute the gradient step but we don’t need to talk about detaching computation graphs etc to be able to state exactly what loss term we are trying to minimize.

As described above, are the ablated and non-ablated networks have the same or different number of parameters?  If different, can we control for this?

**Limitations:**

It would be useful to understand more clearly what if any tradeoffs there are for using progressive loss

**Strengths And Weaknesses:**

The paper is well written and easy to follow.  I like that the idea is not complicated beyond necessity.

I do not think “overthinking” is an appropriate term.  Deterioration of recurrent networks outside their training regime does not correspond to the common sense notion of overthinking.

Given how similar this notion of recall is to existing work, it’s not clear to me that one could consider the architecture itself to be a novel contribution.  The insight that this kind of architecture dramatically helps with this kind of extrapolation is a contribution in its own right, though.

The ablation studies demonstrate the importance of the different components.  However, with the increased input size of $$f_recall$$ over $$f$$, does this change the number of parameters?  A common pitfall for ablation studies in neural networks is to not account for a reduction in the capacity of a network when ablating a component, and it seems like this may be happening here too.

My main worry is that the results here do not generalize beyond the small number of examples shown.  With such a small and simple change, this kind of paper is most convincing with extensive experimentation.

---

> ### Author Response · Authors · 2022-08-01
> **Individual Author Response: nMzj**
>
> We thank the reviewer for their thoughtful and supportive review, and we have addressed their points below:
>
> >Is this correct? If so, I suggest adding an expression like this to the paper. Currently the loss terms are not fully specified.
> >
> We appreciate the clarifying suggestion, and we have added this explanation to Section A.3 of the appendix.
>
> >As described above, are the ablated and non-ablated networks have the same or different number of parameters? If different, can we control for this?
> >
> Due to the additional convolutional layer in the recurrent block, models with recall have a slightly larger number of parameters. We give full details on the architecture of the different models in Section A.1 of the appendix. We agree with the reviewer that controlling for such differences in capacity of models is important. Schwarzchild et al. investigate the effects of depth on the performance of DT nets without recall in their paper titled The Uncanny Similarity of Recurrence and Depth. The benefit of recall **far** outweighs the additional performance boost from extra depth/capacity. We now include a mention of this in appendix A.5.

---

### Meta-Review · Area_Chair_fwM5 · 2022-08-26

**Recommendation:** Accept
**Confidence:** Certain

**Metareview:**

This paper proposes 2 ideas to improve recurrent networks that learn algorithmic tasks, particularly targeting algorithmic extrapolation, i.e. generalizing to much larger instances than seen during training.  The 2 ideas are (1) feed the original input to each recurrent step and (2) a new learning algorithm that encourages the network to predict the output correctly regardless of which recurrent step it is at.  The paper is well written, the ideas are simple but the experiment results clearly demonstrate their usefulness.  All the reviewers are positive about the paper.

I’d recommend accepting this paper.

I do want to also point out a related work “Strong Generalization and Efficiency in Neural Programs” by Li et al., which although different in many places, has a few nice properties similar to the proposed work, including (1) strong generalization, or algorithmic extrapolation where training on small instances learns neural models that can generalize to almost arbitrarily sized data; (2) computation scales with the difficulty of the problem instance rather than tied to the size of the data instance; (3) neural model has no explicit memory of which step of computation they are at and (4) actually learns or “synthesizes algorithms”.

**Award:**

No

---

### Decision · Program_Chairs · 2022-09-14

Accept